# The Potential Transformation Mechanisms of the Marker Components of Schizonepetae Spica and Its Charred Product

**DOI:** 10.3390/molecules25163749

**Published:** 2020-08-17

**Authors:** Xindan Liu, Ying Zhang, Menghua Wu, Zhiguo Ma, Hui Cao

**Affiliations:** Research Center for Traditional Chinese Medicine of Lingnan (Southern China), Jinan University, Guangzhou 510632, China; liuxindan66@stu2018.jnu.edu.cn (X.L.); zyfxwmh@163.com (M.W.); mzg79@hotmail.com (Z.M.)

**Keywords:** Chinese medicinal material, Schizonepetae Spica, Schizonepetae Spica Carbonisata, stir-frying, marker component, transformation mechanism

## Abstract

Schizonepetae Spica (SS) is commonly used for treating colds, fevers, bloody stool and metrorrhagia in China. To treat colds and fevers, traditional Chinese medicine doctors often use raw SS, while to treat bloody stool and metrorrhagia, they usually use Schizonepetae Spica Carbonisata (SSC; raw SS processed by stir-frying until carbonization). However, there have been limited investigations designed to uncover the mechanism of stir-fry processing. In the present study, a method combining gas chromatography-mass spectrometry (GC-MS) and high-performance liquid chromatography (HPLC) was developed for the comprehensive analysis of the chemical profiles of SS and SSC samples. Principal component analysis of the GC-MS data demonstrated that there were 16 significant differences in volatile compounds between the SS and SSC samples. The simultaneous quantification of six nonvolatile compounds was also established based on HPLC, and remarkable differences were found between the two products. These changes were probably responsible for the various pharmacological effects of SS and SSC as well as the observed hepatotoxicity. Finally, the mechanisms could be rationalized by deducing possible reactions involved in the transformation of these marker components. This work reports a new strategy to reveal the chemical transformation of SS during stir-fry processing.

## 1. Introduction

Chinese medicinal materials (CMMs) often have to be processed by using physical or chemical methods before prescription or clinical usage. The aims of processing are to alter the clinical efficacy and/or reduce the toxicity of the CMMs [1]. The classic Chinese medicine literature *Lei Gong Pao Zhi Lun* (Lei Gong Processing Handbook, 500 A.D.) emphasized the importance of processing for the first time as a pharmaceutical technique to fulfill the different requirements of therapy [2]. Modern research has indicated that the processing of CMMs has a large influence on the quality and quantity of chemicals in medicinal materials, which consequently affects the bioactivities, pharmacokinetics, and/or safety of the medicinal materials [3,4,5,6].

Schizonepetae Spica (SS) is the dried spike of *Schizonepeta tenuifolia* Briq. (Chinese Pharmacopoeia, 2015 edition) [7]. It was first recorded in *Shen Nong Ben Cao Jing* (Shen Nong’s herbal classic), a book written 2000 years ago [8]. The herb is commonly used in traditional Chinese medicine (TCM) prescriptions to treat colds, fevers, bloody stool and metrorrhagia [9]. To treat colds and fevers, TCM practitioners often prescribe raw SS, while to treat bloody stool and metrorrhagia, they usually use Schizonepetae Spica Carbonisata (SSC; raw SS processed by stir-frying until carbonization). Pharmacological analyses have shown that SS had three main biological and pharmaceutical properties, including anti-inflammatory [10], antiviral [11,12] and hemostatic activity [13,14,15]. Chemical studies revealed that SS contains volatile oils, flavonoids, organic acids, etc. [16,17,18]. The essential oils accumulated by SS are recognized as the major constituents responsible for its anti-inflammatory and antiviral effects [10,19]. One of the major pharmacological volatile components in SS is pulegone, which is also used as a biomarker for SS in the Chinese pharmacopoeia (2015 edition) [7]. However, evidence has demonstrated that pulegone can cause severe hepatotoxicity [20]. Previous reports have shown that changes in the quantity and quality of the components in the volatile oils of Schizonepetae Herba (SH, the aerial part of *S. tenuifolia* Briq.) and its charred product Schizonepetae Herba Carbonisata (SHC; raw SH processed by stir-frying until carbonization) have been observed: nine new components were formed and eight components disappeared in the volatile oils after processing while the contents of seven original constituents decreased and the concentration of nine other constituents increased [21]. The changes in the contents of some constituents between the raw drugs and processed products are mostly due to the transformation of chemical structures and/or volatilization of volatile compounds during processing. However, the possible mechanism involved in the transformation of marker compounds induced by processing by stir-frying SS has not been fully elucidated. The influence of processing on SS is not only due to the change in volatile oils, and further study on the change in other components is needed. 

In an attempt to uncover the potential mechanism of stir-fry processing on SS, a combination of gas chromatography-mass spectrometry (GC-MS) and high-performance liquid chromatography-diode-array detector (HPLC-DAD) was carried out, thus obtaining an overall characterization for both volatile and nonvolatile compounds. The information obtained from GC-MS was then analyzed by chromatographic fingerprinting to compare the chemical profiles of SS and SSC. Statistical analyses of principal component analysis (PCA) from GC-MS and Student’s two-tailed *t*-tests from HPLC were employed to find marker compounds to differentiate SS from SSC scientifically and reliably. The processing mechanism could be rationalized by deducing possible reactions involved in the transformation of these marker compounds.

## 2. Results and Discussion

### 2.1. Comparison of SS and SSC Volatile Compounds

#### 2.1.1. GC-MS Fingerprint of the Essential Oils from SS and SSC

The essential oils from the SS and SSC samples were extracted by hydrodistillation. The essential oil yield of SS markedly decreased after stir-fry processing, and the essential oil yield from SSC was too low to be detected. It has been reported that stir-frying can dramatically reduce the contents of volatile components from SS through heat volatilization [22]. Nonetheless, the essential oil can still be detected after stir-frying, which supports the theory of CMM processing “raw medicinal herbs processed by stir-frying until carbonized to avoid scorching” [23,24]. All samples were analyzed by GC-MS, and the chromatograms are shown in Figure 1. The correlation coefficient of similarity between each chromatographic profile of SS, SSC and their reference chromatogram, the representative standard fingerprint/chromatogram for a group of chromatograms, was calculated (Table 1). The correlation coefficients from each chromatogram for 11 SS batches were found to be 0.969–0.997, which was in agreement with previous studies [25,26], while those from the 11 batches of SSC were 0.955–0.995 (Table 1). These results demonstrated that the chromatographic fingerprints of SS and SSC were consistent to some extent despite their slightly different chemical compositions, which indicated that the developed processing method for investigating the specific variations between SS and SSC was satisfactory. After stir-fry processing, the GC-MS fingerprints of SS and SSC showed differences in their chromatographic profiling and relative contents (Figure 1). The circled peaks in section Ⅰ and section III had higher contents in SS, whereas section Ⅱ was relatively higher in SSC compared with in SS samples. This phenomenon reminded us that stir-fry processing could cause chemical profile changes in SS.

#### 2.1.2. Identification of Volatile Components from SS and SSC

As shown in Table 2, the compositions of the volatile components of SS and SSC were analyzed. The number of identified peaks in SS and SSC were 39 and 62, accounting for 93.90% and 74.56% of their total volatile content, respectively. The main volatile constituents (>1%) of SS included V18 (*l*-menthone, 13.05%), V19 (menthofuran, 3.18%), V20 (*trans*-5-methyl-2-(1-methylvinyl)cyclohexan-1-one, 2.84%), V25 (2-allyl-4-methylphenol, 1.11%), V26 (pulegone, 58.70%), V37 (3-methyl-6-(1-methylethylidene)cyclohex-2-en-1-one, 3.21%), V48 (caryophyllene, 1.28%) and V83 (linolenic acid, 2.67%), a result that was in accordance with the findings of recent studies [27,28]. The volatile composition and relative contents of SS notably changed after processing by stir-frying, as V18 (*l*-menthone, 1.84%), V19 (menthofuran, 1.46%), V20 (*trans*-5-methyl-2-(1-methylvinyl)cyclohexan-1-one, 1.75%), V26 (pulegone, 27.44%), V35 (2,5,6-trimethylbenzimidazole, 1.01%), V61 (caryophyllene oxide, 1.40%), V72 (3,7,11-trimethyl-1-dodecanol, 1.49%), V73 (neophytadiene, 10.11%), V75 (phytol acetate, 1.58%), V77 (3,7,11,15-tetramethyl-2-hexadecen-1-ol, 3.95%), V80 (methyl palmitate, 1.56%) and V82 (methyl linolenate, 1.28%) were determined to be the main volatile constituents of SSC. In addition, 22 compounds found in SS samples disappeared in SSC samples, while 45 compounds were newly generated and identified in SSC samples. However, in a previous study, it was found that the major compounds limonene and menthone detected in SH were obviously decreased in SHC, whereas the relative contents of another four components, isomenthone, isopulegone, pulegone and piperitone, were higher in SHC than in SH [21]. Different sample preparation methods, different medicinal parts of SH and different sources of SH specimens were most likely the reasons for the variance in the results. In our study, the differences among these variables may lead to efficacy differences in the SS and SSC samples.

#### 2.1.3. Multivariate Statistical Analyses

To further investigate the change in chemical compositions between SS and SSC, the GC-MS data (84 identified peaks) were subjected to PCA analysis. The PCA (*R*^2^*X* = 0.853, *Q*^2^ = 0.788) score plot showed that the 22 samples were obviously separated from the two groups (Figure 2). The first two PCs explained 80.59% of the data variance (PC1 = 72.06% and PC2 = 8.53%); of these, PC1 played a significant role in discriminating SS and SSC samples. SS was located on the negative side of PC1, while SSC was located on the positive side of PC1. The corresponding loading plot of PC1 (Figure 3) was used to find the components responsible for the separation between SS and SSC. The signals giving a positive effect in PC1 demonstrated that the corresponding ingredients were higher in SSC than in SS. In contrast, the signals with negative values indicated that the level of related components was higher in SS. The signals of major constituents V18 (*l*-menthone), V19 (menthofuran), V20 (*trans*-5-methyl-2-(1-methylvinyl)cyclohexan-1-one), V25 (2-allyl-4-methylphenol), V26 (pulegone), V37 (3-methyl-6-(1-methylethylidene)cyclohex-2-en-1-one), V48 (caryophyllene) and V83 (linolenic acid) gave a negative contribution to PC1. The signals with positive PC1 values included major components V35 (2,5,6-trimethylbenzimidazole), V61 (caryophyllene oxide), V72 (3,7,11-trimethyl-1-dodecanol), V73 (neophytadiene), V75 (phytol acetate), V77 (3,7,11,15-tetramethyl-2-hexadecen-1-ol), V80 (methyl palmitate) and V82 (methyl linolenate). The volatile components of SS have been recognized as the major constituents responsible for its biological effects. For example, pulegone, which is known for its pleasant odor, analgesia, and anti-inflammatory and antiviral properties [19,29], is a chemical indicator of SS in the Chinese Pharmacopoeia (2015 edition) [7]. *L*-Menthone also presents analgesia and antiviral effects [19]. Caryophyllene is a functional cannabinoid receptor type 2 agonist [30]. Menthofuran is widely used in flavorings and fragrances [31]. On the other hand, neophytadiene is a dominant metabolite in *Urtica dioica* L., a folk medicine that is commonly used as a hemostatic agent [32]. Therefore, these compounds might be the discriminant marker compounds when distinguishing SS from SSC, which are characterized by different medicinal properties.

#### 2.1.4. Possible Mechanisms Involved in the Transformation of Volatile Compounds between SS and SSC

Building on the chemical marker analysis described above and the extensive organic synthesis knowledge accumulated by literature data [33,34,35,36,37,38], the possible mechanisms involved in the transformation of volatile compounds during stir-frying SS are demonstrated in Figure 4. For example, the significant decline in pulegone content in Table 2 revealed that the chemical reaction involved might generate an epoxidation reaction between the original pulegone (V26) and oxygen during stir-frying [33]. According to previous studies [19,29], pulegone has been demonstrated to have good biological activity in terms of anti-inflammatory and antiviral effects. However, evidence has proven that pulegone and its oxidative product *p*-cresol can cause severe hepatotoxicity [27]. Thus, the epoxidation reaction may occur in the process of preparing SSC by stir-frying raw SS, allowing SSC to contain less toxic pulegone. On the other hand, it has been found that *β*-Caryophyllene exhibits potent anti-inflammatory activity [39]. Stir-fry processing may also reduce the SS’s anti-inflammatory property because of the changing of *β*-Caryophyllene (V48) to caryophyllene oxide (V61) and 10,10-dimethyl-2,6-dimethylenebicyclo-[7.2.0]undecan-5*β*-ol (V66) [34,35]. Meanwhile, neophytadiene (V73) seemed to be generated from the dehydration reaction of 3,7,11,15-tetramethyl-2-hexadecen-1-ol (V77), a product that was generated from fitone (V74) by hydroxylation during stir-frying [35,36]. Neophytadiene is a major constituent in *U. dioica*, a folk medicine that is commonly used as a hemostatic agent [32]. Hence, the traditional use of SSC is in line with the observed increase in the amount of neophytadiene with plausible hemostatic activity. Likewise, linolenic acid (V83) was assumed to be transformed into methyl linolenate (V82) by esterification reaction during stir-frying [38]. Previous study has reported that linolenic acid shows bioactivities such as anti-inflammatory and antioxidant effects [40]. Therefore, it might be also related to the disappearance or significant decrease of linolenic acid (Table 2) and efficacy changing under the stir-fry processing of SS.

### 2.2. Comparison of the Nonvolatile Components of SS and SSC

#### 2.2.1. Determination of Nonvolatile Compound Contents of SS and SSC

Representative HPLC chromatograms of the SS and SSC samples are shown in Figure 5. The six nonvolatile compounds were identified using reference standards of luteolin-7-*O*-*β*-d-glucoside, apigenin-7-*O*-*β*-d-glucoside, hesperidin, rosmarinic acid, luteolin and apigenin, as well as comparing with literature [41,42,43]. Comparison of the chromatograms of SS and SSC revealed that stir-fry processing mainly caused quantitative changes among the nonvolatile compounds. The simultaneous determination of the six compounds in SS and SSC was analyzed (Table 3). Stir-fry processing significantly (*p* < 0.001) decreased the total nonvolatile compound contents of SS from 5.29 ± 0.42 mg/g to 1.56 ± 0.06 mg/g. Compared with SS, the contents of four major ingredients (luteolin-7-*O*-*β*-d-glucoside, apigenin-7-*O*-*β*-d-glucoside, hesperidin and rosmarinic acid) decreased significantly (*p* < 0.001), whereas another two main components (luteolin and apigenin) increased significantly in SSC samples (*p* < 0.001). However, the intensity of some relatively polar compounds with shorter retention times on reversed-phase chromatography that were detected in SSC was markedly higher than those in SS (Figure 5). The identities of these compounds and the mechanisms involved in their transformation need to be further studied. Previous studies have demonstrated that stir-fry processing causes a significant increase in the total flavonoid and tannin contents of SH and SS [44,45,46]. Total flavonoids [47], tannins [48] and carbon dots [49] were proven to be active principles with hemostatic properties in SHC. In our research, the contents of total flavonoids (luteolin-7-*O*-*β*-d-glucoside, apigenin-7-*O*-*β*-d-glucoside, hesperidin, luteolin and apigenin) decreased remarkably after the SS samples were processed. A reasonable explanation for this result is that few flavonoids were detected in the ethanol extract of SSC.

#### 2.2.2. Possible Mechanisms Involved in the Transformation of the Nonvolatile Compounds between SS and SSC

In the present study, the changes in the contents of luteolin-7-*O*-*β*-d-glucoside and apigenin-7-*O*-*β*-d-glucoside probably contributed to the increases in luteolin and apigenin, respectively. The possible mechanisms involved in the transformation of these compounds are assumed in Figure 6. The results were consistent with previous reports in that glucoside could be hydrolyzed to aglycone under various conditions, including high temperature during stir-fry processing [43,50,51]. In addition, luteolin-7-*O*-*β*-d-glucoside has been documented to possess significant anti-inflammatory and antiviral effects [52,53]. Luteolin has been shown to be a potent hemostatic drug candidate [54]. Therefore, the increased contents of luteolin might result from the degradation of the glucosidic bond of luteolin-7-*O*-*β*-d-glucoside, which could be a main mechanism of stir-fry processing in SS.

## 3. Materials and Methods

### 3.1. Materials and Reagents

SS and SSC samples came from the same path (Table 1). SSC was processed in accordance with the Chinese Pharmacopoeia (2015 edition) [7]: dried SS slices (10–15 mm in length) were put in a pan and stir-fried until the surface became black and the interior became brown to avoid scorching. The identities of SS and SSC were confirmed by Dr. Ying Zhang, Jinan University, P. R. China. Voucher specimens were deposited at the Research Center for Traditional Chinese Medicine of Lingnan (Guangzhou, Southern China), Jinan University.

Ethyl acetate (analytical grade) was purchased from Aladdin (Aladdin, Shanghai, China). Methanol (HPLC grade), acetonitrile (HPLC grade) and formic acid (HPLC grade) were purchased from Fisher Scientific (Fair Lawn, NJ, USA). The chemical standards luteolin-7-*O*-*β*-d-glucoside, hesperidin, rosmarinic acid, luteolin and apigenin were purchased from Chengdu Ruifensi Biotechnology (Sichuan, China). apigenin-7-*O*-*β*-d-glucoside was purchased from Tianjun Biotechnology (Guangzhou, China). The purity of the compounds was greater than 95% as determined by HPLC. A mix of alkane analogues (GC grade, purity > 97%), which was used as an internal quality standard for GC-MS analysis, was purchased from o2si (Charleston, SC, USA). Deionized water was obtained by passing distilled water through a Milli-Q purification system (Millipore, Bedford, MA, USA). All other reagents were of analytical grade.

### 3.2. Sample Preparation

For GC-MS analysis, the steam distillation method was chosen according to the Chinese Pharmacopoeia (2015 edition) for the extraction of essential oils [7]. All samples were smashed and filtered through a 24-mesh sieve. The dried powder (50 g) was accurately weighed and transferred to a 1000 mL flask and soaked in 300 mL of redistilled water for 1 h. Redistilled water was added from the top of the volatile oil determination apparatus until the water spilled into the flask, and 2 mL of ethyl acetate was added to the water layer. Then, the essential oils were extracted by water distillation for 4 h. The volatile oil was separated from the water layer and leached into the ethyl acetate layer, and then the ethyl acetate layer was dried over anhydrous sodium sulfate for GC-MS analysis. The anhydrous essential oils were stored in dark glass vials at −20 °C until use.

For HPLC analysis, 2 g of the pulverized sample (filtered through a 24-mesh sieve) was weighed accurately and macerated in 20 mL of 75% ethanol. The sample was then extracted for 60 min by hot reflux extraction in a water bath. The supernatant of the extracts was filtered through a 0.45-μm membrane before injection. The samples were stored in a refrigerator at 4 °C until use.

### 3.3. Standard Solution Preparation

A mixed standard stock solution containing the 6 reference compounds was prepared in methanol. Working standard solutions were prepared by diluting the mixed standard stock solution with methanol to give different concentrations within the following ranges for calibration curves: luteolin-7-*O*-*β*-d-glucoside, 0.796–159.2 μg/mL; apigenin-7-*O*-*β*-d-glucoside, 0.48–48 μg/mL; hesperidin, 2.084–625.2 μg/mL; rosmarinic acid, 0.924–277.2 μg/mL, luteolin, 0.472–141.6 μg/mL and apigenin, 0.4–120 μg/mL. All standard solutions were filtered through a 0.45-μm membrane before injection. The solutions were stored in a refrigerator at 4 °C before analysis.

### 3.4. Apparatus and Chromatographic Conditions

The GC-MS instrument used was an Agilent 7890B GC system coupled to an Agilent 7000C GC/MS Triple Quad mass spectrometer (Agilent, Santa Clara, CA, USA). Initial chromatographic separations of 1 μL of sample were performed on a 15 m × 250 μm i.d. × 0.25 μm film thickness HP-5 (Agilent) capillary column with a He flow rate of 1.0 mL/min and an injection port temperature of 250 °C with a split ratio of 1:10. The oven temperature ramp was 3 min at 50 °C, then 10 °C/min to 90 °C where the temperature was held for 5 min, then ramped at the rate of 10 °C/min to 160 °C where the temperature was maintained for 10 min, then a 20 °C/min ramp to 260 °C where the temperature was held for 3 min. The detector was operated at an ionization energy of 70 eV, and the m/z values were recorded in the range 50–600 amu with a scan rate of 3.6 scan/s and a solvent delay of 3 min. Components were identified using the National Institute of Standards and Technology (NIST) 2.2L Mass Spectra Database containing approximately 189,000 compounds, as well as comparing with literature [27,55,56,57].

HPLC analysis was performed on an UltiMate 3000 liquid chromatography system (Thermo Scientific, Bremen, Germany). Chromatographic separation was performed with an ACE Excel 5 C18 column (4.6 mm × 250 mm, 5 μm; ACE, Scotland, UK), and the column temperature was maintained at 30 °C. The mobile phase was 0.1% formic acid aqueous solution (A) and acetonitrile (B) with a gradient program as follows: 0–20 min, linear gradient from 8–20% B; 20–30 min, isocratic elution at 20% B; 30–50 min, linear gradient from 20–30% B; 50–60 min, linear gradient from 30–8% B. The flow rate was 1 mL/min, and the injection volume was 10 μL. The detection wavelength was set at 283 nm.

### 3.5. Data Analysis

The GC-MS fingerprint was performed by the professional software Similarity Evaluation System for Chromatographic Fingerprint of Traditional Chinese Medicine (Version 2004 A) composed by the Chinese Pharmacopoeia Committee. PCA was performed by SIMCA-P version 11.5 software (Umetrics, Umea, Sweden). Statistical significance was assessed by Student’s two-tailed *t*-tests with GraphPad Prism v. 5.0 software (San Diego, CA, USA). Values of *p* < 0.05, *p* < 0.01 or *p* < 0.001 were considered to be statistically significant.

## 4. Conclusions

This work reported, for the first time, the application of the combination of GC-MS and HPLC methods coupled with a strategy of corresponding data processing and statistical analysis to qualitatively and quantitatively distinguish SS and SSC and to identify marker compounds with significantly changed structures or contents during stir-fry processing. The GC-MS comparative analysis results of SS and SSC showed that 16 major constituents had a large contribution to the discrimination. HPLC analyses revealed that stir-fry processing remarkably increased the contents of two main ingredients but significantly reduced the contents of another four major constituents from SS. The change in the type and amount of these marker components is probably responsible for the different functions and pharmacological effects of SS and SSC as well as the observed hepatotoxicity. Finally, we speculated how some of these marker chemical changes occurred upon stir-fry processing. Hence, the potential mechanisms of stir-fry processing could be justified. The proposed strategy provided new clues for the investigation of the stir-frying-induced chemical transformation of SS and provided useful references for understanding the potential mechanisms of other processing methods.

## Figures and Tables

**Figure 1 molecules-25-03749-f001:**
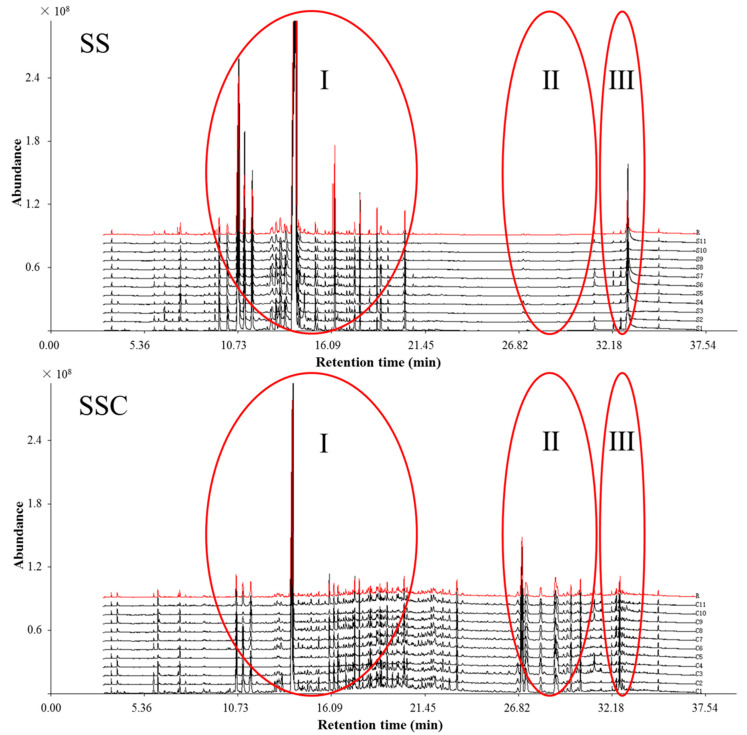
GC-MS fingerprints of Schizonepetae Spica (SS) and Schizonepetae Spica Carbonisata (SSC) samples and their reference chromatograms (R).

**Figure 2 molecules-25-03749-f002:**
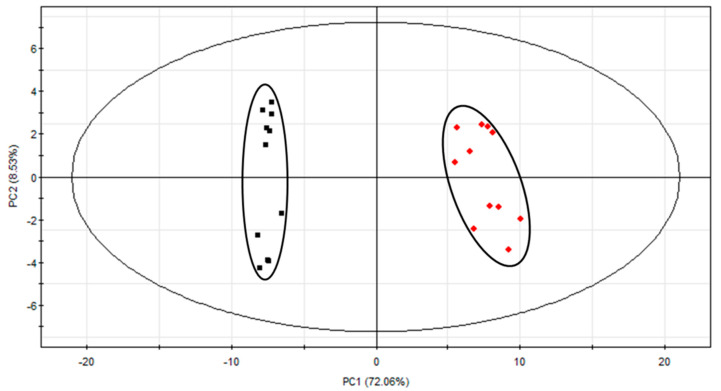
Principal component analysis (PCA) score plots based on gas chromatography-mass spectrometry (GC-MS) data of SS (black) and SSC (red).

**Figure 3 molecules-25-03749-f003:**
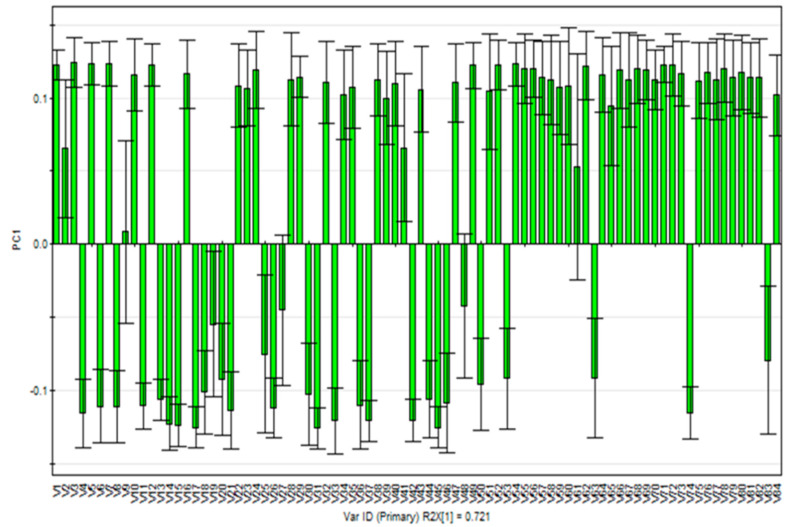
PC1 loading plot of the PCA results obtained from the GC-MS spectra.

**Figure 4 molecules-25-03749-f004:**
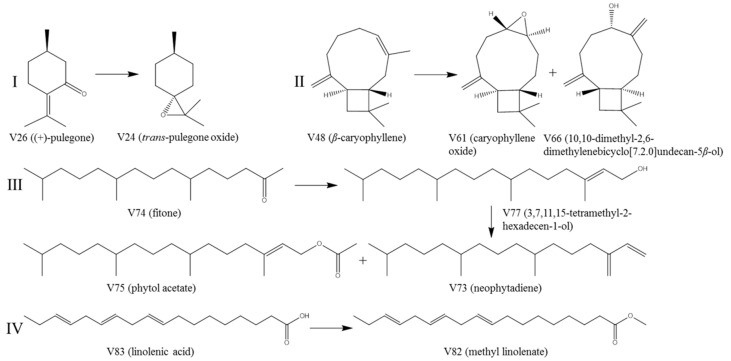
Possible mechanisms involved in the transformation of some main volatile components during stir-fry processing of SS.

**Figure 5 molecules-25-03749-f005:**
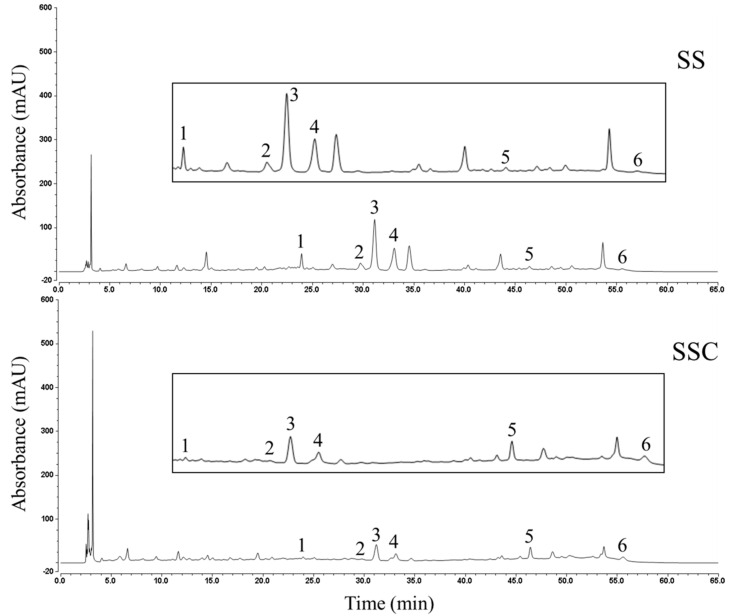
Representative HPLC chromatograms of SS and SSC samples. 1, luteolin-7-*O*-*β*-d-glucoside; 2, apigenin-7-*O*-*β*-d-glucoside; 3, hesperidin; 4, rosmarinic acid; 5, luteolin; and 6, apigenin.

**Figure 6 molecules-25-03749-f006:**
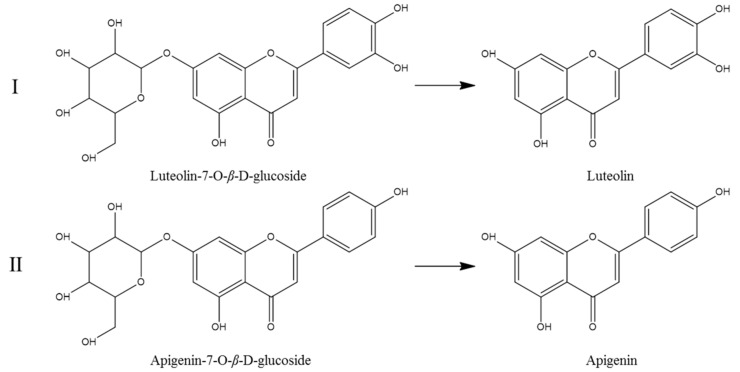
Possible mechanisms involved in the transformation of some main nonvolatile components during the stir-fry processing of SS.

**Table 1 molecules-25-03749-t001:** Details of the 11 batches SS and SSC samples and their fingerprint similarities.

No.	Voucher Specimen	Place of Collection	Date of Collection	Similarity	
				SS	SSC
1	YP132581301	Henan province	29 November 2018	0.982	0.974
2	YP132581401	Henan province	17 January 2019	0.969	0.995
3	YP132581501	Guangdong province	8 January 2019	0.997	0.972
4	YP132581601	Hubei province	9 January 2019	0.995	0.962
5	YP132581701	Hubei province	10 January 2019	0.984	0.955
6	YP132581801	Shanxi province	9 January 2019	0.990	0.989
7	YP132580101	Hebei province	7 June 2018	0.989	0.978
8	YP132582301	Hebei province	9 January 2019	0.994	0.976
9	YP132582601	Jiangsu province	11 January 2019	0.991	0.995
10	YP132582901	Jiangxi province	11 January 2019	0.983	0.984
11	YP132583001	Jiangxi province	12 January 2019	0.994	0.977

**Table 2 molecules-25-03749-t002:** Volatile compounds and their relative contents in SS and SSC.

No.	Compound	Molecular Formula	RI	CAS	Similarity Indices	Relative Content (%)	
						SS	SSC
V1	Butyl acetate	C_6_H_12_O_2_	818.86	123-86-4	81.07%	-	0.24 ± 0.06
V2	(*R*)-(+)-3-Methylcyclohexanone	C_7_H_12_O	952.53	13368-65-5	82.43%	0.10 ± 0.04	0.23 ± 0.16
V3	Benzaldehyde	C_7_H_6_O	965.57	100-52-7	93.42%	-	0.52 ± 0.10
V4	1-Octen-3-ol	C_8_H_16_O	983.39	3391-86-4	78.46%	0.18 ± 0.06	-
V5	(1*R*,2*S*,5*S*)-2-Methyl-5-(3-oxoprop-1-en-2-yl)cyclopentane-1-carbaldehyde	C_10_H_14_O_2_	993.51	5951-57-5	85.63%	-	0.07 ± 0.02
V6	1,3,8-*p*-Menthatriene	C_10_H_14_	1005.59	18368-95-1	82.08%	0.04 ± 0.02	-
V7	2-Ethyl-1,4-dimethylbenzene	C_10_H_14_	1022.65	1758-88-9	91.27%	-	0.15 ± 0.03
V8	*E*,*E*-2,6-Dimethyl-1,3,5,7-octatetraene	C_10_H_14_	1023.77	460-01-5	88.83%	0.08 ± 0.03	-
V9	*D*-Limonene	C_10_H_16_	1026.40	5989-27-5	95.44%	0.38 ± 0.35	0.42 ± 0.17
V10	3,5-Dimethyl-2-cyclohexen-1-one	C_8_H_12_O	1042.58	1123-09-7	83.79%	-	0.15 ± 0.05
V11	Benzeneacetaldehyde	C_8_H_8_O	1043.37	122-78-1	91.68%	0.11 ± 0.05	-
V12	1-Ethenyl-3,5-dimethyl-benzene	C_10_H_12_	1088.07	5379-20-4	88.24%	-	0.15 ± 0.04
V13	2,5-Dimethylstyrene	C_10_H_12_	1089.74	2039-89-6	80.65%	0.07 ± 0.03	-
V14	Linalool	C_10_H_18_O	1100.42	78-70-6	83.43%	0.10 ± 0.02	-
V15	*trans*-1-Methyl-4-(1-methylvinyl)cyclohex-2-en-1-ol	C_10_H_16_O	1118.90	7212-40-0	89.17%	0.90 ± 0.17	-
V16	1-Phenyl-1-butene	C_10_H_12_	1122.33	824-90-8	83.12%	-	0.06 ± 0.02
V17	*cis*-*p*-Mentha-2,8-dien-1-ol	C_10_H_16_O	1133.26	3886-78-0	91.38%	0.81 ± 0.12	-
V18	*l*-Menthone	C_10_H_18_O	1148.46	14073-97-3	90.28%	13.05 ± 6.27	1.84 ± 1.75
V19	Menthofuran	C_10_H_14_O	1159.04	494-90-6	81.68%	3.18 ± 2.34	1.46 ± 1.11
V20	*trans*-5-Methyl-2-(1-methylvinyl)cyclohexan-1-one	C_10_H_16_O	1171.24	29606-79-9	88.11%	2.84 ± 0.61	1.75 ± 0.56
V21	(−)-*cis*-Isopiperitenol	C_10_H_16_O	1202.38	96555-02-1	91.15%	0.73 ± 0.28	-
V22	3-Methyl-2-(2-methyl-2-butenyl)-furan	C_10_H_14_O	1208.98	15186-51-3	86.28%	-	0.25 ± 0.12
V23	4,7-Dimethylbenzofuran	C_10_H_10_O	1214.35	28715-26-6	84.75%	-	0.43 ± 0.23
V24	*trans*-Pulegone oxide	C_10_H_16_O_2_	1218.24	13080-28-9	79.80%	-	0.40 ± 0.10
V25	2-Allyl-4-methylphenol	C_10_H_12_O	1222.89	6628-06-4	92.37%	1.11 ± 0.56	0.43 ± 0.37
V26	(+)-Pulegone	C_10_H_16_O	1245.58	89-82-7	91.91%	58.70 ± 11.53	27.44 ± 7.85
V27	Piperitone	C_10_H_16_O	1258.51	89-81-6	86.72%	0.35 ± 0.19	0.22 ± 0.14
V28	5-Isopropenyl-2-methyl-2-cyclohexen-1-yl pivalate	C_15_H_24_O_2_	1276.52	1000124-59-2	79.10%	-	0.20 ± 0.07
V29	4-Ethyl-2-methoxyphenol	C_9_H_12_O_2_	1283.03	2785-89-9	83.34%	-	0.36 ± 0.14
V30	(7-Hydroxy-3,3-dimethyl-4-oxo-7-vinylbicyclo [3.2.0]hept-1-yl)acetaldehyde	C_13_H_18_O_3_	1290.48	1000156-78-3	79.14%	0.40 ± 0.23	-
V31	(1*S*,3*S*,5*S*)-1-Isopropyl-4-methylenebicyclo[3 .1.0]hexan-3-yl acetate	C_12_H_18_O_2_	1294.44	139757-62-3	83.38%	0.20 ± 0.03	-
V32	Thymol	C_10_H_14_O	1297.47	89-83-8	84.05%	-	0.30 ± 0.15
V33	Carveol	C_10_H_16_O	1312.88	99-48-9	85.39%	0.13 ± 0.03	-
V34	(4-Methoxymethoxy-hex-5-ynylidene)-cyclohexane	C_14_H_22_O_2_	1315.74	1000186-16-6	80.43%	-	0.15 ± 0.08
V35	2,5,6-Trimethylbenzimidazole	C_10_H_12_N_2_	1328.07	3363-56-2	88.88%	-	1.01 ± 0.61
V36	Carvyl acetate	C_12_H_18_O_2_	1335.28	97-42-7	84.83%	0.10 ± 0.04	-
V37	3-Methyl-6-(1-methylethylidene)cyclohex-2-en-1-one	C_10_H_14_O	1342.11	491-09-8	90.88%	3.21 ± 0.55	0.84 ± 0.28
V38	1,1,5-Trimethyl-1,2-dihydronaphthalene	C_13_H_16_	1354.07	1000357-25-8	92.67%	-	0.61 ± 0.24
V39	1,2,3,4-Tetrahydro-1,1,6-trimethylnaphthalene	C_13_H_18_	1357.25	475-03-6	83.46%	-	0.17 ± 0.10
V40	*α*-Ethyl-4-methoxybenzenemethanol	C_10_H_14_O_2_	1371.02	5349-60-0	87.85%	-	0.17 ± 0.08
V41	*α*-Copaene	C_15_H_24_	1376.65	1000360-33-0	90.42%	0.11 ± 0.04	0.27 ± 0.19
V42	(−)-*β*-Bourbonene	C_15_H_24_	1385.02	5208-59-3	92.16%	0.13 ± 0.03	-
V43	Falcarinol	C_17_H_24_O	1390.30	21852-80-2	88.33%	-	0.12 ± 0.06
V44	*β*-Elemene	C_15_H_24_	1393.37	515-13-9	79.54%	0.08 ± 0.04	-
V45	Jasmone	C_11_H_16_O	1399.03	488-10-8	87.93%	0.07 ± 0.01	-
V46	2-[(2*Z*)-2-Buten-1-yl]-4-hydroxy-3-methyl-2-cyclopenten-1-one	C_10_H_14_O_2_	1401.35	17190-74-8	85.22%	0.39 ± 0.18	-
V47	1-Methyl-4-[(2-methyl-3-butyn-2-yl)oxy]benzene	C_12_H_14_O	1403.18	82719-54-8	87.66%	-	0.90 ± 0.50
V48	*β-*Caryophyllene	C_15_H_24_	1420.14	87-44-5	94.76%	1.28 ± 0.78	0.89 ± 0.25
V49	(3*β*,5*α*)-2-Methylenecholestan-3-ol	C_28_H_48_O	1455.41	22599-96-8	80.11%	-	0.60 ± 0.17
V50	Humulene	C_15_H_24_	1458.11	6753-98-6	86.35%	0.15 ± 0.10	-
V51	*β*-Guaiene	C_15_H_24_	1479.26	88-84-6	85.83%	-	0.11 ± 0.07
V52	4-(2,4,4-Trimethyl-cyclohexa-1,5-dienyl)-but-3-en-2-one	C_13_H_18_O	1484.88	1000187-51-9	84.20%	-	0.52 ± 0.16
V53	Germacrene D	C_15_H_24_	1485.14	23986-74-5	90.97%	0.80 ± 0.56	-
V54	1-Chlorooctadecane	C_18_H_37_Cl	1496.85	3386-33-2	84.04%	-	0.61 ± 0.17
V55	Bicyclo[4.1.0]heptan-2-ol, 2-Hydroxy-2,6-dimethyl-1-[(1*E*)-3-methyl-1,3-butadien-1-yl]bicyclo[4.1.0]hept-3-yl acetate	C_16_H_24_O_3_	1507.78	1000196-25-1	83.45%	-	0.25 ± 0.08
V56	(+)-*δ*-Cadinene	C_15_H_24_	1523.45	483-76-1	91.52%	0.13 ± 0.05	0.96 ± 0.30
V57	4,5,9,10-Dehydroisolongifolene	C_15_H_20_	1542.96	156747-45-4	88.25%	-	0.51 ± 0.25
V58	3-(2-Methyl-propenyl)-1*H*-indene	C_13_H_14_	1559.02	1000187-78-5	82.46%	-	0.53 ± 0.27
V59	4,4-Dimethyl-3-(3-methylbut-3-enylidene)-2-methylenebicyclo[4.1.0]heptane	C_15_H_22_	1563.49	79718-83-5	82.83%	-	0.49 ± 0.28
V60	(−)-Spathulenol	C_15_H_24_O	1579.60	77171-55-2	81.56%	0.14 ± 0.04	0.48 ± 0.19
V61	Caryophyllene oxide	C_15_H_24_O	1584.80	1139-30-6	85.95%	0.87 ± 0.66	1.40 ± 0.66
V62	Geranyl isovalerate	C_15_H_26_O_2_	1595.20	109-20-6	81.47%	-	0.56 ± 0.17
V63	(1*R*,3*E*,7*E*,11*R*)-1,5,5,8-Tetramethyl-12-oxabicyclo[9.1.0]dodeca-3,7-diene	C_15_H_24_O	1612.71	19888-34-7	83.83%	0.06 ± 0.05	-
V64	(8*S*)-1-Methyl-4-isopropyl-7,8-dihydroxy-spiro[tricyclo[4.4.0.0(5,9)]-decane-10,2’-oxirane]	C_15_H_24_O_3_	1613.74	1000193-77-2	81.05%	-	0.15 ± 0.07
V65	*α*-Santonin	C_15_H_18_O_3_	1628.24	481-06-1	81.28%	-	0.19 ± 0.16
V66	10,10-Dimethyl-2,6-dimethylenebicyclo-[7.2.0]undecan-5*β*-ol	C_15_H_24_O	1637.26	19431-80-2	84.47%	-	0.29 ± 0.11
V67	2-[4-methyl-6-(2,6,6-trimethylcyclohex-1-enyl)hexa-1,3,5-trienyl]cyclohex-1-en-1-carboxaldehyde	C_23_H_32_O	1646.83	1000216-09-2	79.63%	-	0.32 ± 0.16
V68	13-Heptadecyn-1-ol	C_17_H_32_O	1662.98	56554-77-9	81.41%	-	0.58 ± 0.19
V69	*cis*-5,8,11,14,17-Eicosapentaenoic acid	C_20_H_30_O_2_	1668.82	10417-94-4	82.21%	-	0.95 ± 0.29
V70	(1*R*,7*S*,*E*)-7-Isopropyl-4,10-dimethylenecyclodec-5-enol	C_15_H_24_O	1687.47	81968-62-9	81.29%	0.03 ± 0.02	0.39 ± 0.15
V71	3,4’-Diethyl-1,1’-biphenyl	C_16_H_18_	1708.74	61141-66-0	88.07%	-	0.45 ± 0.09
V72	3,7,11-Trimethyl-1-dodecanol	C_15_H_32_O	1723.32	6750-34-1	80.42%	-	1.49 ± 0.38
V73	Neophytadiene	C_20_H_38_	1834.08	504-96-1	90.35%	-	10.11 ± 3.20
V74	Fitone	C_18_H_36_O	1842.16	502-69-2	87.33%	0.05 ± 0.02	-
V75	Phytol acetate	C_22_H_42_O_2_	1860.42	1000375-01-4	85.59%	-	1.58 ± 0.60
V76	2,4,7,14-Tetramethyl-4-vinyl-tricyclo[5.4.3.0(1,8)]tetradecan-6-ol	C_20_H_34_O	1865.55	1000193-31-2	84.74%	-	0.26 ± 0.09
V77	3,7,11,15-Tetramethyl-2-hexadecen-1-ol	C_20_H_40_O	1879.81	102608-53-7	85.05%	-	3.95 ± 1.53
V78	Hexadecanenitrile	C_16_H_31_N	1895.78	629-79-8	81.46%	-	0.63 ± 0.19
V79	1-Heptatriacotanol	C_37_H_76_O	1914.84	105794-58-9	81.06%	-	0.21 ± 0.08
V80	Methyl palmitate	C_17_H_34_O_2_	1931.28	112-39-0	88.36%	-	1.56 ± 0.48
V81	Methyl linoleate	C_19_H_34_O_2_	2091.82	112-63-0	92.97%	-	0.70 ± 0.26
V82	Methyl linolenate	C_19_H_32_O_2_	2097.76	301-00-8	90.66%	0.08 ± 0.06	1.28 ± 0.45
V83	Linolenic acid	C_18_H_30_O_2_	2149.50	463-40-1	94.44%	2.67 ± 2.41	-
V84	2,2’-Methylenebis(6-tert-butyl-4-methylphenol)	C_23_H_32_O_2_	2428.07	119-47-1	87.78%	0.11 ± 0.02	0.28 ± 0.10

Relative content (%) in the last two columns represents the mean ± SD (*n* = 11). RI, retention index. CAS, Chemical Abstracts Service. Similarity indices obtained from direct searching with the National Institute of Standards and Technology (NIST) MS database.

**Table 3 molecules-25-03749-t003:** Nonvolatile compound contents of SS and SSC.

Contents of Nonvolatile Compounds (mg/g)							
	Luteolin-7-*O*-*β*-d-glucoside	Apigenin-7-*O*-*β*-d-glucoside	Hesperidin	Rosmarinic Acid	Luteolin	Apigenin	Total
SS	0.40 ± 0.06	0.25 ± 0.09	2.97 ± 0.99	1.62 ± 0.67	0.04 ± 0.04	0.01 ± 0.01	5.29 ± 0.42
SSC	0.05 ± 0.01 ***	0.02 ± 0.01 ***	0.64 ± 0.16 ***	0.40 ± 0.11 ***	0.32 ± 0.05 ***	0.13 ± 0.03 ***	1.56 ± 0.06 ***

Data are presented as the mean ± SD (*n* = 11). *** *p* < 0.001 vs. SS samples. The total nonvolatile compound contents were calculated as the sum of the contents of the 6 individual nonvolatile compounds.

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
