# Peer review of "The Potential Transformation Mechanisms of the Marker Components of Schizonepetae Spica and Its Charred Product"

_molecules, 2020, doi:10.3390/molecules25163749_

Round 1

Reviewer 1 Report

Manuscript presents a study on chromatographic profiling and comparison of constituents between fresh and processed herbal drug from Traditional Chinese Medicine. Mechanisms of chemical transformations during heat processing are described. The discussion and conclusions are adequately supported by the results. I recommend publication after authors correct the minor issues outlined below.

Comments:

Keywords: “mechanism” is a very general term and should be given more specifically as a keyword.

Introduction

Line 46: “oils” in SS – fatty or volatile (essential) oils? Please specify.

Lines 55-57: Certainly the decrease in concentration during processing by heat could also be due to volatilization, not only transformation; especially if the active constituents are volatile.

Lines 62-63: Which detection mode was used in HPLC?

Results & Discussion

Table 2: Please be careful about proper writing of chemical names (same in the text), i.e. p- (para-) and elemental annotations (e.g. 1H-) should be in italics. It would also be good to add an index or percent of similarity (depends on the software) between the unknown compound mass spectrum and a spectrum from mass spectral library.

Section 2.1.4 and Figure 4: Fig. 4 shows 4 transformation mechanisms, but the authors chose to explain only two of them (and the possible effects on the medicinal properties of the remedy). Other two transformations should be explained as well, or else omitted from the figure.

Lines 180-181: How were the compounds identified in the first place? Were these constituents known from some previous study – please give the reference?

Figure 5: Only the part of chromatograms with relevant compounds should be shown and appropriatelly enlarged to clearly show the peaks.

Line 189: “relatively polar peaks” is incorrect. Compounds are polar, but not peaks.

Materials and Methods

Line 236: “n-alkane” is not a compound. Please specifiy which alkane or a mix of alkane analogues was used.

Lines 252-253: Extraction by ethanol by hot reflux (probably at 100 oC) could already cause some decomposition of glycosides or other chemical transformations, which would affect the content of compounds. Did authors check this possibility?

Reviewer 2 Report

A paper entitled “The potential transformation mechanisms of the marker components of Schizonepetae Spica and its charred product” is submitted to Molecules for further reviewing and publication. This paper described the volatile- and nonvolatile compounds from the SS and SSC. It is very interesting for the study on TCM. Paper is well written and the conclusion on the possible mechanism in the transformation of the nonvolatile compounds between SS and SSC is valuable for further study. I recommended that this submission is acceptable for publication after revision.

Minor comments

  1. The key words “gas chromatography-mass spectrometry” “high-performance liquid chromatography” and “mechanism” are pointless. The authors should be provided the key words related to the research.
  2. The font size for Table 1 should be reuniform as that of other Tables shown in the text.
